# Method and Device Based on Multiscan for Measuring the Geometric Parameters of Objects

**Michael Yurievich Alies** [1], **Yuriy Konstantinovich Shelkovnikov** [1], **Milan Sága** [2], **Milan Vaško** [2,*], **Ivan Kuric** [3], **Evgeny Yurievich Shelkovnikov** [1], **Aleksandr Ivanovich Korshunov** [1] and **Anastasia Alekseevna Meteleva** [4]

1   Federal State Budgetary Institution of Science, Udmurt Federal Research Center of the Ural Branch of the Russian Academy of Sciences, Institute of Mechanics, T. Baramzinoy str. No. 34, 426067 Izhevsk, Russia; director@udman.ru (M.Y.A.); yushelk@mail.ru (Y.K.S.); evshelk@mail.ru (E.Y.S.); kai@udman.ru (A.I.K.)
2   Department of Applied Mechanics, Faculty of Mechanical Engineering, University of Žilina, 010 26 Žilina, Slovakia; milan.saga@fstroj.uniza.sk
3   Department of Automation and Production Systems, Faculty of Mechanical Engineering, University of Žilina, 010 26 Žilina, Slovakia; ivan.kuric@fstroj.uniza.sk
4   Faculty of Computer Engineering, Kalashnikov Izhevsk State Technical University, Studencheskaya str. No. 7, 426069 Izhevsk, Russia; meteleva-nami@rambler.ru
*   Correspondence: milan.vasko@fstroj.uniza.sk; Tel.: +421-41-513-2981

**Abstract:** The article deals with the issues of improving the accuracy of measurements of the geometric parameters of objects by optoelectronic systems, based on a television multiscan. A mathematical model of a multiscan with scanistor activation is developed, expressions for its integral output current and video signal are obtained, and the mechanism of their formation is investigated. An expression for the video signal is obtained that reflects the dual nature of the discrete–continuous multiscan structure: the video signal can have a discrete (pulse) or analog (continuous) form, depending on the step voltage between the photodiode cells of the multiscan. A Vernier discrete–analog method for measuring the parameters of the light zone on a multiscan is proposed, in which in order to increase the accuracy of the measurements, the location of the video pulse is determined relative to the neighboring reference pulses of a rigid geometric raster due to the slope of the discrete structure of the multiscan. It is established that the Vernier method enables one to make precision measurements of the coordinates, dimensions, and movements of the light zones by an overlay on a video raster of reference pulses from cells—a uniform sequence of Vernier pulses with a recurrence interval, followed by determining the number of the Vernier pulse that coincides with the raster pulse. An optoelectronic device based on a discrete–continuous multiscan, implemented on the basis of the proposed Vernier method of measuring the coordinates of the light zones, which has a high sensitivity to movement, is characteristic of continuous structures, and has increased stability and linearity of the coordinate characteristics typical for discrete structures, is developed.

**Keywords:** multiscan; measurement; photodiode cell; discrete–continuous structure; video signal; Vernier method

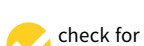

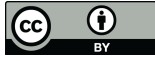

## 1. Introduction

The development of television, robotics, automatic control and monitoring systems, image recognition, and the creation of visual receptors for artificial intelligence systems is directly related to the development of highly sensitive and high-speed vision systems [1–5]. Television scanistor structures (solid scanistor, multi-element multiscan) are designed to convert the spatial distribution of illumination into electrical signals used for monitoring, processing, and transmission of optical information in technical vision systems, especially for automatic contactless monitoring and measurement of coordinates, dimensions, and movements of various objects [6–13]. It should be noted that the multiscan (due to its discrete–continuous structure) provides better metrological characteristics than

the scanistor, due to the low level of dark currents, stability, and high symmetry of the volt–ampere characteristics of the photodiode cells, uniform photosensitivity, as well as due to the absence of longitudinal conductivity and a higher linearity of the coordinate characteristic [14,15]. The multi-element multiscan enables complex miniaturization and improvement of the metrological characteristics of radio-electronic control devices, processing, and transmission of optical information [16–18]. The multi-element multiscan is structurally designed as an integrated circuit and has three lines of discrete point p–n junctions, $D1_i$, $D2_i$, and $D3_i$ (Figure 1), and two longitudinally distributed solid resistive dividers, $R1$ and $R2$. Diodes $D1_i$ and $D2_i$ of the extreme lines are switching, whereas diode $D3_i$ of the middle line is photosensitive. Solid dividers have a resistance of 5–20 kΩ and are provided with pins 1–2 and 3–4; the "anodes" of the photodiodes are connected to a common low-resistance output bus (pin 5). Switching diodes can be both photosensitive and shielded from light, and the number of cells $N = 400$, the step of the cell structure $S = 50$ μm, and the length of the multiscan 20 mm [12].

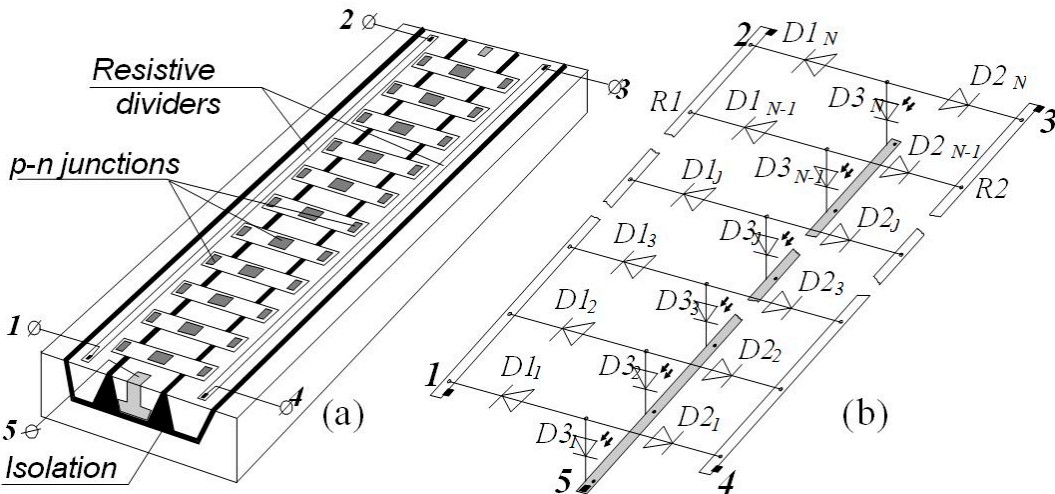

**Figure 1.** (**a**) General view of the multiscan; (**b**) its equivalent scheme.

Multiscan-based optoelectronic systems are characterized by a high accuracy, speed, reliability, and ease of technical implementation [19–22]. However, there are still little-studied issues affecting the accuracy of the linear measurements, related to the properties of the multi-element multiscan as a coordinate-sensitive photo-converter. The purpose of this work is to improve the accuracy of measuring the geometric parameters of objects using multiscan [23–27].

## 2. Features of Multiscan Operation According to the "Scanistor" Activation Scheme

A discrete–solid multiscan, in contrast to a solid scanistor, has a more complex structure and a larger number of pins, so different switching schemes are possible. There are several basic schemes allowing multiscan: "butterfly", "blind spot", "photopotentiometer", and "scanistor" enabling, according to the most commonly used traditional scheme (similar to the scheme for solid scanistor) [6]; and a modified scheme, with extended functionality (allows you to obtain a signal with the current value of the total light flux at the same time as the video signal). In [19], for a modified scheme (based on a discrete analog method for video signal selection and processing [11]), to improve the accuracy of the measurements of the coordinates and dimensions of the light zones, the temporal coordination of the information video signal between the neighboring reference video pulses of the discrete cells of the structure, with precisely known spatial coordinates, is used.

Analysis of multiscan modes and schemes has shown that the traditional "scanistor" circuit (Figure 2), with the simplest multiscan switching and low voltage survey in the analog mode, provides highly accurate measurements with the necessary geometric pa-

rameters of the object's coordinates, dimensions, and movement, as well as controlling the luminance distribution along the multiscan photosensitive surface. In this most commonly used traditional "scanistor" switching scheme, multiscan continuous resistive dividers are connected in parallel, supplied with a constant voltage from an OVS offset voltage source, to one of the pins of which the SVG sawtooth voltage generator of the opposite polarity is connected. Switching of the next discrete unit cell of the multiscan is performed at the moment when the polling voltage reaches the voltage level on the OVS cell. The photodiode of the cell switches from the closed to the open position, switching the diodes from the open to closed position. If the next interrogated cell is illuminated, its photodiode and switching diodes produce an elementary output current $j_L$ corresponding to the illumination. The total output current $I_L$ from the unibus of the multiscan photodiodes is converted using the CVC current–voltage converter into a voltage. After its differentiation by the differentiating amplifier DA, a video signal $V$ is generated in proportion to the illumination distribution along the photosensitive surface of the multiscan.

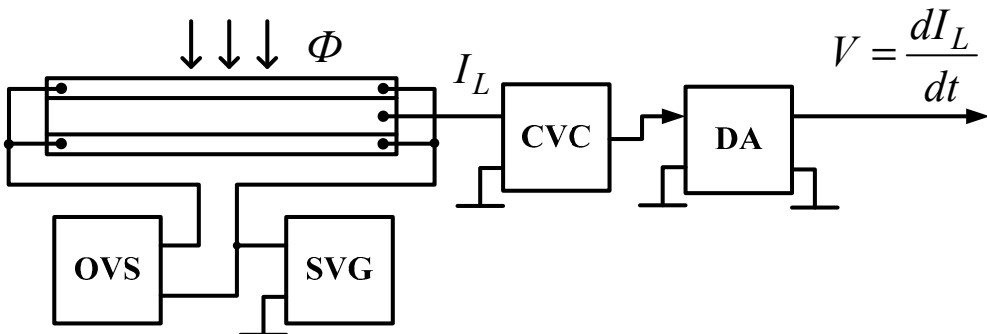

**Figure 2.** Enabling a multiscan using the traditional "scanistor" scheme.

To improve the metrological characteristics of multiscan-based measuring devices, it is important to explore the possibility of sharing the positive properties of its discrete continuous structure (high geometric accuracy of the photodiode cells' placement and high coordinate sensitivity to the movement of the light zone of the information video signal).

### 3. Mathematical Model of a Multiscan in a Scanistor Enabling

The mathematical model of a multiscan, which adequately describes its operation, is a system of equations for the voltages and currents of branches of an equivalent multiscan circuit in a scanistor enabling (Figure 3).

$$E_{ep1} = E_c + U_{be1} + U_{bc}; \tag{1}$$

$$E_{ep2} = E_c + U_{be2} + U_{bc}; \tag{2}$$

$$j_{be1} = j_{s1}\left(e^{\alpha U_{be1}} - 1\right) - j_{be1}^f; \tag{3}$$

$$j_{be2} = j_{s2}\left(e^{\alpha U_{be2}} - 1\right) - j_{be2}^f; \tag{4}$$

$$j_{bc} = -j_{s3}\left(e^{\alpha U_{bc}} - 1\right) + j_{bc}^f; \tag{5}$$

$$I_L = 2I_{bc}, \tag{6}$$

where $E_{ep1} = E_0 \frac{x_0}{l_1}$; $E_{ep2} = E_0 \frac{x_0}{l_2}$—potential of dividing layers 1, 2 at the polling point $x_0$; $E_c = E_0 \frac{t_0}{T}$—the value of the sawtooth voltage at the time of the survey $t_0$; $T$—sawtooth period; $U_{be1}$, $U_{be2}$—base-emitter junction voltage 1, 2; $U_{bc}$—base-collector junction voltage; $j_{be1}$, $j_{be2}$—total currents through base-emitter junctions 1, 2; $j_{be1}^T$, $j_{be2}^T$—dark saturation currents of the base-emitter junctions 1, 2; $j_{be1}^f$, $j_{be2}^f$—increments of base-emitter junction saturation currents 1, 2 under lighting; $j_{bc}$—total current through the base-collector junction;

$j_{bc}^f$—increment of the saturation current of the base-collector junction in lighting; $j_{bc}^T$—dark saturation base-collector junction current; $R_L$—load resistance; $l_1$, $l_2$—lengths of dividing layers 1, 2; $\alpha = A\frac{KT}{q}$; $K$—Boltzmann constant; $q$—electron charge; $T$—Kelvin temperature; and $A$—coefficient reflecting the degree of "non-ideality" of the p–n junction.

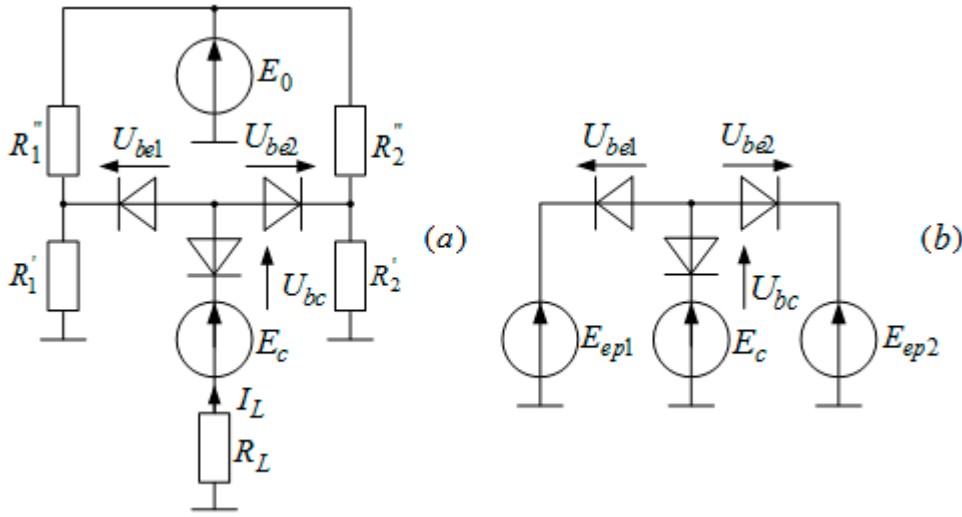

**Figure 3.** Equivalent multiscan scheme for an elementary cross-section of its structure in the scanistor mode: (**a**) full scheme; (**b**) with the replacement of voltage dividers $R_1$ and $R_2$, and the source of the offset voltage OVS by the voltage sources $E_{ep1}$ and $E_{ep2}$ at the time of the multiscan survey.

For multiscan in the scheme (Figure 2) $l_1 = l_2 = l$; $E_{ep1} = E_{ep2} = E_{ep}$; $U_{be1} = U_{be2} = U_{be}$; $j_{be1}^T = j_{be2}^T = j_{be}^T$; $j_{be1}^f = j_{be2}^f = j_{be}^f$; $j_{be1} = j_{be2} = j_{be}$; and Equations (1)–(6) can be rewritten in the forms

$$E_{ep} = E_c + U_{be} + U_{bc}, \tag{7}$$

$$j_{be} = j_s\left(e^{\alpha U_{be}} - 1\right) - j_{be}^f, \tag{8}$$

$$j_{bc} = -j_s\left(e^{\alpha U_{bc}} - 1\right) + j_{bc}^f, \tag{9}$$

$$j_L = 2j_{bc}. \tag{10}$$

From Equations (8) and (9) it follows:

$$U_{be} = \frac{1}{\alpha}\ln\frac{j_{be} + j_{be}^s + j_{be}^f}{j_{be}^s}; \tag{11}$$

$$U_{bc} = \frac{1}{\alpha}\ln\frac{j_{be}^s + j_{be}^f - j_{be}}{j_{be}^s}. \tag{12}$$

Substituting Equations (11) and (12) into (7), and solving the latter with respect to $j_{bc}$, gives an expression for the output current $j_L$ of the unit cell:

$$j_L = 2\left(j_s\frac{exp\,\alpha(E_e - E_c) - 1}{exp\,\alpha(E_e - E_c) + 1} + j_{bc}^f\frac{exp\,\alpha(E_e - E_c)}{exp\,\alpha(E_e - E_c) + 1} - j_{be}^f\frac{1}{exp\,\alpha(E_e - E_c) + 1}\right) \tag{13}$$

In order to obtain the total current $I_L$ of a discrete–continuous multiscan structure by loading, it is appropriate to express it as integral data, especially since in conventional traditional schemes for multiscan use (at low bias voltages of dividing buses and interrogation voltages) its structure is discrete and the output current has a continuous (analog) character. This is due to the fact that due to diffusion of the carriers injected by p–n junctions of one cell, they contribute to the currents flowing through the other multiscan cells. The

condition of the analog mode is a value such as $\Delta U \leq 2U_0$ of the step voltage at which at least two photodiode cells are in the current switching state. In this case, the bell-shaped video signals from the adjacent switched discrete cells (in which the photodiodes of the cells are separated by longitudinal gaps where the photocurrents and dark currents are not formed) partially overlap each other in the non-photosensitive area between them, and when the step voltage $\Delta U = U_0$ is applied, they become indistinguishable, and the overall video signal acquires a trapezoidal shape. Therefore, this current $i\,(x_n, t)$ of the nth cell can be replaced by the linear current density along the multiscan equal to $i(x, t) = \frac{i(x_n,t)}{S}$ (where $S$ is the structure step). Then, the total current $I_L$ with the coordinate $x_0$ at the time of the survey $t_0$ can be interpreted as the output continuous analog current of the conditional solid structure, $I_L(x, t) = \int j_L(x, t)$.

Total current through $R_L$:

$$
\begin{aligned}
I_L &= b \cdot \int_0^l 2\left[ j_s \frac{exp\ \alpha(E_e - E_c) - 1}{exp\ \alpha(E_e - E_c) + 1} + j_{bc}^f \frac{exp\ \alpha(E_e - E_c)}{exp\ \alpha(E_e - E_c) + 1} - j_{be}^f \frac{1}{exp\ \alpha(E_e - E_c) + 1} \right] dx \\
&= \frac{2l \cdot b \cdot U_0}{E_0} \left( j_f + j_s \right) \cdot \{ 3\ ln[exp\ \alpha(E_e - E_c) + 1] - \alpha(E_e - E_c) \}
\end{aligned}
\tag{14}
$$

An analog video signal $V$ from a multiscan is formed when the total current $I_L$ is differentiated:

$$
\begin{aligned}
V = L\frac{dI_L}{dt} &= 2Lb \Bigg\{ \int_0^l \frac{d\left[ j_s \frac{exp\ \alpha(E_e - E_c) - 1}{exp\ \alpha(E_e - E_c) + 1} \right]}{dt} dx + \int_0^l \frac{d\left[ j_{bc}^f \frac{exp\ \alpha(E_e - E_c)}{exp\ \alpha(E_e - E_c) + 1} \right]}{dt} dx \\
&+ \int_0^l \frac{d\left[ -\frac{j_{be}^f}{exp\ \alpha(E_e - E_c) + 1} \right]}{dt} dx \Bigg\} = \frac{2L \cdot b \cdot l}{T} \left( 2j_s + j_{bc}^f + j_{be}^f \right) \left[ \frac{1}{exp\ \alpha(E_e - E_c) + 1} \right]_0^1,
\end{aligned}
\tag{15}
$$

where $L$—coefficient depending on the method of differentiation $I_L$; $b$—multiscan width.

The video signal for the case when a light zone with a width from $x_1$ to $x_2$ is projected onto the multiscan is described by the expression

$$
V = \frac{4L \cdot b \cdot l}{T} j_s \left[ \frac{1}{exp\ \alpha(E_e - E_c) + 1} \right]_0^l + \frac{2L \cdot b \cdot l}{T} \left( j_{bc}^f + j_{be}^f \right) \left[ \frac{1}{exp\ \alpha(E_e - E_c) + 1} \right]_{x_1}^{x_2}.
\tag{16}
$$

## 4. The Mechanism for Generating a Video Signal from a Discrete-Continuous Multiscan Structure

Equation (16) reflects the dual nature of a discrete–continuous multiscan structure: the video signal $V$ can have a discrete (pulse) or analog (continuous) form, depending on the step voltage $\Delta U = \frac{E_0}{N}$ between the multiscan cells. When polling a multiscan with a sawtooth voltage, a slight change in current is simultaneously formed in all its cells. However, for the interrogated cell in the zone of its switching (under the condition $\Delta U \geq 4U_0$ corresponding to discrete mode), the current magnitude and its direction rapidly changed, which creates a stepped form of the total current in the output bus. If all the multiscan cells are fully illuminated, then the differentiation of its successive output current produces a video signal in the form of a rigid geometric raster of N video pulses. Each of the pulses, by its location in the raster, uniquely corresponds to the technologically specified exact position of this cell in the discrete structure of the multiscan. The video signal (16) is for this case written in the form of Equation (17), where the addresses of the illuminated cells are given by the coordinates of the beginning $x_{b_i}^c$ and end $x_{e_i}^c$ of the photodiode sections of these cells:

$$
V = \frac{4L \cdot b \cdot l}{T} j_s \left[ \frac{1}{exp\ \alpha(E_e - E_c) + 1} \right]_0^l + \frac{2L \cdot b \cdot l}{T} \left( j_{bc}^f + j_{be}^f \right) \cdot \sum_{i=1} \left[ \frac{1}{exp\,\alpha(E_e - E_c) + 1} \right]_{x_{b_i}^c}^{x_{e_i}^c}
\tag{17}
$$

Figure 4 shows the modeling results in Mathcad of the change in the output current of the multiscan (according to Equation (14)) and the video signal (according to Equation (17)) for its first ten cells, of which cells Nos. 2–9 are uniformly illuminated with a decrease in the bias voltage $E_0 = 80; 40; 20B$ (which corresponds to the step voltage $\Delta U = 2U_0 \cdot (0.05B)$, $4U_0 \cdot (0.1B)$, and $8U_0 \cdot (0.2B)$).

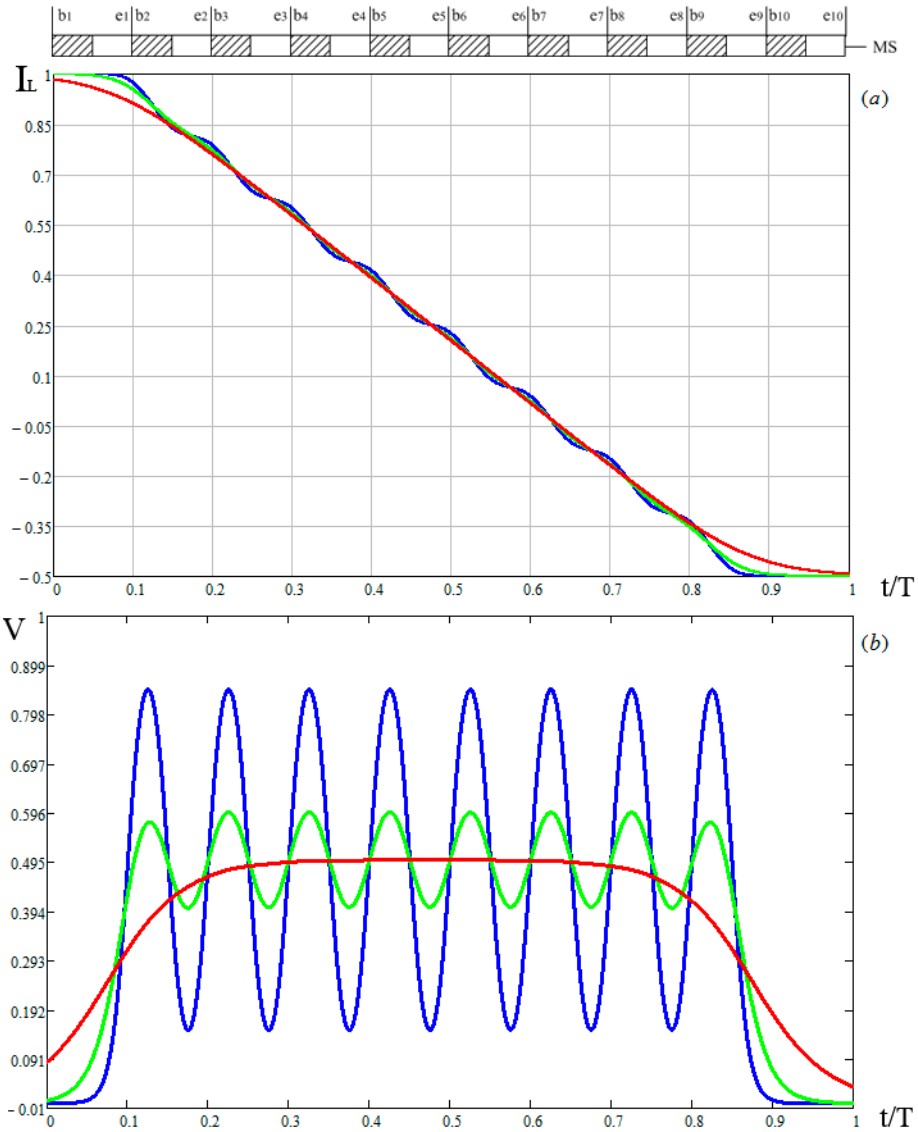

**Figure 4.** Change in the output current and video signal of a multiscan with a decrease in the bias voltage $E_0$: (**a**) Dependence of variables $I_L$-$t$; (**b**) dependence of variables $V$-$t$.

Analysis of the current and video signal graphs (Figure 4) showed the following:

1.  At voltage $E_0 = 80 \cdot B \cdot (\Delta U \geq 4U_0 \geq 0.1B)$, from each photodiode cell, a photodiode (inclined) corresponding to the transitional section of its I–V characteristic and non-photosensitive (horizontal) sections of the current graphs corresponding to the dividing bus are formed. The bell-shaped video signals are also formed with non-photosensitive (horizontal) sections between the adjacent video signals from each photodiode section of the cells, which corresponds to a discrete multiscan operation mode. It should be noted that for an ideal I–V, the cells' transitional (inclined) sections of the current graphs become rectangular (described by the Heaviside function), the bell-shaped video signals are in the form of delta functions, and the sections between them are not photosensitive.

2. Reducing the voltage $E_0$ to $40B \cdot (\Delta U = 4U_0 = 0.1B)$ reduces the non-photosensitive (horizontal) part of the current graphs to zero, which are then transformed into a straight line. The bell-shaped video signals from neighboring cells are also merged into the total video signal of a trapezoidal multiscan, which corresponds to the boundary between the discrete and analog modes.

3. With a further decrease in $E_0$ to the $20B \cdot (\Delta U \le 2U_0 \le 0.05B)$, the current graphs do not change and are in the form of straight lines, the slopes of which are determined by the slope of the transition section of the I–V of the cell. The video signals also remain trapezoidal and their amplitude is proportional to the illumination of the photosensitive surface of the multiscan at the polling point, which corresponds to the analog mode of the multiscan operation.

The condition for generating the video information signal in the analog mode is the potential distribution on the dividing bus, in which at any time the sawtooth polling voltage in the current switching state is at least two photodiode cells. This ensures a high sensitivity of the temporal position of the characteristic points of the video signal to the minimum movement of the light zone on the photosensitive surface of the multiscan. The required bias voltage on the dividing bus is determined from the condition $E_0 \le 4U_0 \cdot N \le \Delta U \cdot N$.

It should be noted that, in the analog mode $\Delta U \le 2U_0$, the bell-shaped video signals from the neighboring cells are combined and the trapezoidal video signal of the multiscan described by Equation (16) (Figure 4) becomes completely analogous to the video signal from a continuous scanistor [8].

This ensures high sensitivity of the temporary position of the characteristic points of the video signal to the minimum movement of the light zone (LZ) on the photosensitive surface of the multiscan (the characteristic points of the video signal are the inflection points of the leading and trailing edges of the video signal from the wide and narrow LZ, and the maximum point of the video signal from the narrow LZ). The time coordinate of the position of the narrow LZ ($(x_2 - x_1) < 2 \cdot \Delta x_s$, where $\Delta x_s$ is the multiscan cell current switching zone) is uniquely determined by the moment when the first derivative of the video signal passes zero. The temporal coordinate of the middle of the wide LZ ($(x_2 - x_1) > 2 \cdot \Delta x_s$) can be determined by the half-sum of the time instants $t_m = (t_1 + t_2)/2$ that transitions through zero of the second derivative of the video signal. The width of both the wide and narrow LZ can be determined by the difference in time instants $t_w = (t_2 - t_1)$ of the zero crossings of the second derivative of the video signal. The width and coordinates of the narrow and wide LZ can also be determined by the video signal without its differentiation, if you measure the time moments corresponding to the transitions of the video signal through half its maximum level. These time points correspond to the inflection points of the leading and trailing edges of the video signal, which are at half the maximum level according to Equation (16) (for which at the moment of measuring the coordinates of the inflection points $E_e = E_c$; $(exp\ \alpha\ (E_e - E_c) + 1)^{-1} = 0.5$).

## 5. Discrete–Analog Vernier Method and a Device Based on It for Determining the Coordinates of Objects

The high coordinate sensitivity of the multiscan information video signal when determining the temporary location of its characteristic points in analog mode, as well as the technologically specified accuracy of the placement of the photodiode cells in the discrete multiscan structure can be used to increase the accuracy of measuring the geometric parameters of objects. For this purpose, in the proposed Vernier discrete–analog method for measuring the LZ parameters and the location of the information video pulse is determined not in relation to the beginning or end of the sawtooth scan voltage period, but in relation to the adjacent reference video pulses of a rigid geometric raster, due to the topology of the discrete multiscan structure. The Vernier method allows precise measurements of the coordinates, sizes, and movements of the light zones by superimposing on the video images from ($n \cdot 10$) reference pulses from the cells' uniform sequence of Vernier pulses, with a

repetition period $T_{vp} = (n \cdot 10 - 1) \cdot \frac{T_{rp}}{n \cdot 10}$ (where $n$ is an integer), as well as in the subsequent determination of the number of the Vernier pulse coinciding with the raster impulse.

The Vernier method combines the linearity and stability over time of the photoelectric conversion characteristics (due to the stiffness of the geometric raster of the reference pulses) with high sensitivity to the displacement of the LZ of the continuous structures (due to the possibility of measurement of the coordinate of the information video pulse between the neighboring raster pulses). This eliminates the time instability and the distortion of the linearity of the coordinate characteristic when changing the operating parameters of the measuring devices based on this method.

Consider a device implemented on the basis of the proposed nonius method for measuring the coordinates of light zones in which a Vernier pulse generator is launched when the information video pulse appears, and the location of the information pulse with respect to the adjacent raster pulses is determined at the moment of the combination of the raster and nonius pulses as the sum of the coordinates of the last raster pulse before the information video pulse and the distance corresponding to the time interval from the moment the video pulse appears to the moment of the combination of the Vernier and raster pulses.

The functional diagram of the Vernier device for measuring the coordinates of objects is shown in Figure 5 (where Ob—controlled object; MSx, MSy—multiscans along the X, Y axes; LS1, LS2—light sources; BSC—beam splitting cube; OSx, OSy—optycal systems; BVS—bias voltage source; BDV—block of deployment voltages; BSP—block of synchronizing pulses; VSEU—video signals extraction unit; VPSU—video pulse selection unit; RPEU—reference pulse extraction unit; VPG—Vernier pulse generator; PDU—pulse delay unit; T1, T2, T3—triggers; AND1, AND 2, AND 3, AND 4—"And" schemes; PC1, PC2, PC3—pulse counters; CFU—coordinate forming unit; ECy—electronic channel for measurement coordinate Y; and MP—microprocessor). The measurement of the coordinates of the object Ob is performed during two periods $T$ of the unfolding sawtooth voltage. In the first period T1, the image Ob illuminated by the light source LS1 is projected in the form of a light zone onto the MSx multiscan, which is interrogated by a low voltage from the BDV unit (Figure 6a). The block VSEU of the video signal extraction selects the video signal, and the block VPSU of the video pulse selection generates a pulse $U_{t_x}$ corresponding to the time coordinate $t_x$ of the LZ, which enters the block PDU of the pulse delay for the time delay for a period $T$.

In the second period T2, the pulse synchronization block BSP turns off the LS1 and turns on the light source LS2 to uniformly irradiate the entire multiscan, and the BDV block polls the multiscan MSx with a high voltage. The RPEU block of the $U_{RP}$ reference pulse extraction forms from the video signal (Figure 6b), and from the VSEU output a hard raster of pulses $U_{RP}$ (Figure 6c), the temporary position of which corresponds to the geometric coordinates of the multiscan photodiode cells.

When a delayed video pulse $T'$ occurs (Figure 6c), the Vernier pulse generator VPG is started and T2 is triggered at the same time, the output signal of which allows the raster pulses (RP) to pass with a period $T_{vp} = (n \cdot 10 - 1) \cdot \frac{T_{rp}}{n \cdot 10}$ (where $n$ is integer) to the first input AND3, and to the second input, of which the raster pulses arrive at the RPEU. Over time, the interval between the adjacent raster and Vernier pulses (VP) decreases and at its minimum value the pulses begin to overlap. The AND2 circuit is triggered and a pulse of coincidence of raster and Vernier pulses is formed. The Vernier channel is introduced for more precise time coordination of the information video pulse, not in relation to the beginning of the scanning period, but in the interval between the raster time instants for which the corresponding exact spatial coordinates on the multiscan are known. The spatial coordinate $x_i$ of the LZ can be defined as $x_i = n_i \cdot S + \frac{\Delta \tau \cdot S}{\tau_i}$ (where $n_i$ is the pulse number of the photodiode cell until the delayed video pulse $U'_{t_3}$ appears; $\Delta \tau$ is the interval from the moment of appearance $U_{n_i}$ to the pulse $U'_{t_x}$; and $\tau_x$ is the interval between $U_{n_i}$ and $U_{n_{i+1}}$).

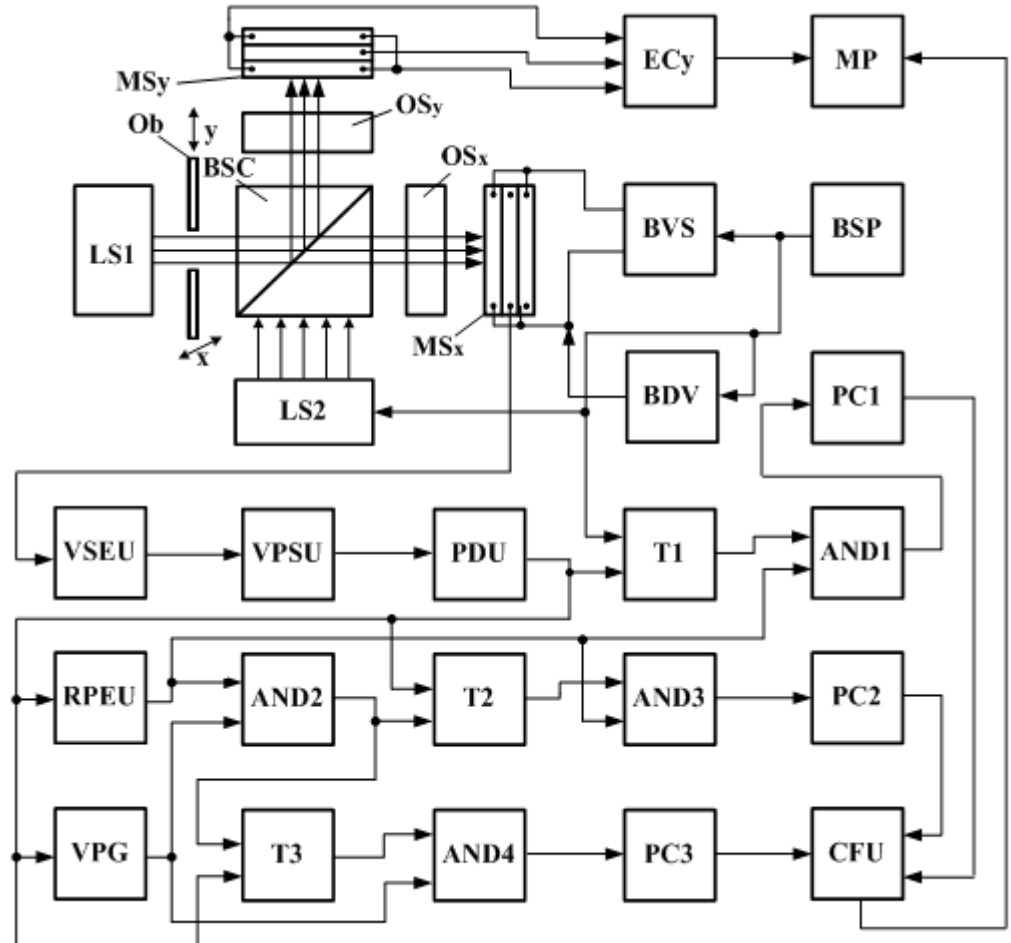

**Figure 5.** Functional diagram of a device for measuring the coordinates of objects.

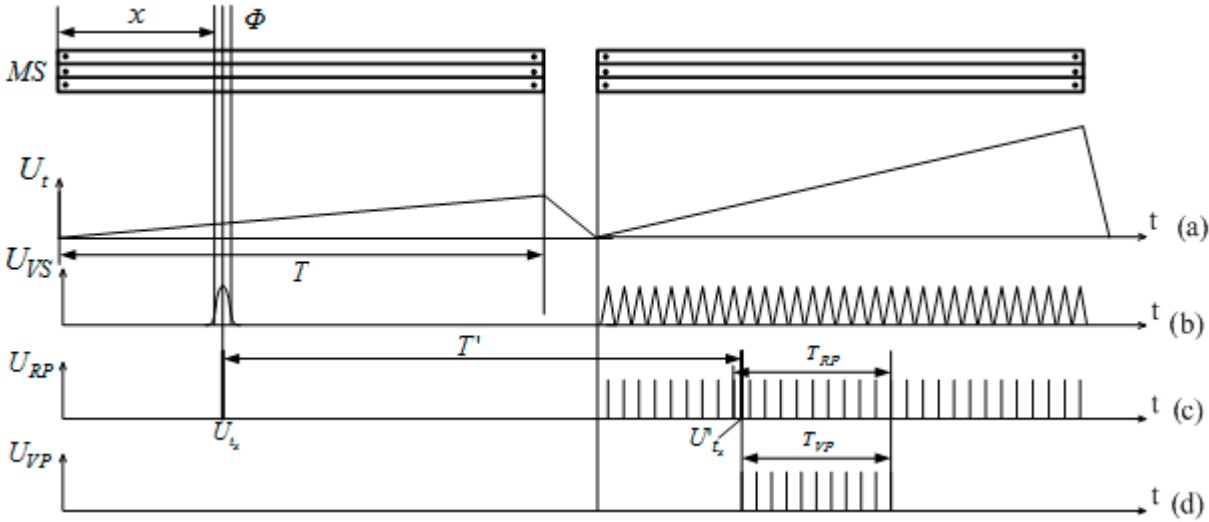

**Figure 6.** Timing diagrams of the device for measuring the coordinates of objects: (**a**) Dependence of variables $U_t$-$t$; (**b**) dependence of variables $U_{VS}$-$t$; (**c**) dependence of variables $U_{RP}$-$t$; (**d**) dependence of variables $U_{VP}$-$t$.

The raster strobe at the output T2 of the duration from the last $U_{RP}$ at the output of the AND1 circuit until the output pulse combining the raster and Vernier pulses of the AND2 is applied to the first input of the AND3 circuit, the second output of which is supplied with raster pulses made of the RPEU.

Their number from the moment of the last output $U_{RP}$ of the AND1 circuit to the moment of the combining pulse appears is counted by the PC2 counter and transmitted to the CFU block to generate the x-coordinate of the object for calculating the time interval using the formula $T_{RP} = n_{RP} \cdot \Delta T_{RP}$ (Figure 6c).

The Vernier strobe at the output of the T3 trigger, lasting from the beginning of the information video signal until the output, combines pulses CP of RP, and the VP pulses of the AND2 circuit arrives at the first input of the AND4 circuit, the second input of which is supplied with Vernier pulses from the VPG. The number $T_{VP}$ from the moment the video pulse appears to the moment of occurrence; the combining CP pulse is calculated using the PC3 counter and transmitted to the CFU to calculate the time interval $T_{VP}$ (according to the formula $T_{VP} = n_{VP} \cdot \Delta T_{VP}$ (Figure 6d)), for calculating the measured interval $\Delta \tau = T_{RP} - T_{VP}$ between the last RP at the output of the AND1 circuit and the moment the video pulse appears.

The advantage of the proposed device is the combination of a high sensitivity to displacements inherent in continuous structures with increased stability and linearity of the coordinate characteristic contained in discrete structures.

## 6. Conclusions

1. It is shown that, for the most commonly used traditional switching circuits of the multiscan "scanistor", it is appropriate to study its discrete–continuous structure in order to achieve the potential metrological characteristics of the circuit due to its high sensitivity to small displacements of the light zone on the multiscan and the ability to generate rigid geometric raster video pulses from its discrete photodiode cells.
2. A mathematical model of a multiscan with a scanistor inclusion was developed, expressions for its integral output current were obtained, and the mechanism of their formation was studied.
3. A Vernier discrete analog method for measuring the parameters of the light zone on a multiscan is proposed, in which in order to increase the accuracy of the measurements, the location of the information video pulse is determined relative to the adjacent reference pulses of a rigid geometric raster due to the topology of the discrete structure of the multiscan.
4. It was established that the Vernier method allows precise measurements of the coordinates, sizes, and movements of the light zones by superposition on the video raster of the $n \cdot 10$ reference pulses from the cells—a uniform sequence of Vernier pulses with a repetition period $T_{vp} = (n \cdot 10 - 1) \cdot \frac{T_{rp}}{n \cdot 10}$ with the subsequent determination of the Vernier pulse that coincides with the raster pulse.
5. An optical–electronic device based on a discrete–continuous multiscan was developed and implemented on the basis of the proposed Vernier method for measuring the coordinates of the light zones, which has a high sensitivity to displacements inherent in continuous structures and increased stability due to the linearity of the coordinate characteristics of discrete structures.

**Author Contributions:** Conceptualization, M.Y.A., Y.K.S., A.I.K., M.S. and M.V.; methodology, M.Y.A., Y.K.S., E.Y.S. and A.I.K.; software, A.A.M., A.I.K. and I.K.; validation, M.Y.A., Y.K.S., E.Y.S. and A.I.K.; investigation, M.Y.A., Y.K.S., E.Y.S. and A.I.K.; writing—original draft preparation, M.Y.A., Y.K.S. and M.V.; writing—review and editing, M.Y.A., Y.K.S., M.V. and M.S.; supervision, M.Y.A. All authors have read and agreed to the published version of the manuscript.

**Funding:** The work has been supported by the Cultural and Educational Grant Agency of the Ministry of Education of the Slovak Republic in project KEGA No. 001ŽU-4/2020 and project KEGA No. 037ŽU-4/2018.

**Conflicts of Interest:** The authors declare no conflict of interest.

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
