# Peer review of "Method and Device Based on Multiscan for Measuring the Geometric Parameters of Objects"

_processes, doi:10.3390/pr9010024_

Round 1
Reviewer 1 Report
Paper is good represented. Introduction provides sufficient background and are included most relevant references. The used methods are adequately described. The conclusions are supported by the clearly presented results.
Author Response
Dear Reviewer,
Thank You for the review of our submitted manuscript. Individual reviewers had various comments on some parts of the article (even contradictory). We have tried to make changes and improvements in the article based on the compromise and your comments.
Comments and Suggestions for Authors
Paper is good represented. Introduction provides sufficient background and are included most relevant references. The used methods are adequately described. The conclusions are supported by the clearly presented results.

Reviewer 2 Report
1. Photocurrent is composed of many different linear superposition states, but how to determine the specific states.
2. If there is only one current-to-voltage converter, how to achieve parallel connection of each circuit by layout to in the system
3. Image recognition is easier by Fourier transform, why not use it.
4. In Figure 4, it can mark the data line, otherwise it is more difficult to determine what each data line is.
5. The photodetector senses the light intensity. I want to know how it scans the object when the light intensity of the object is the same. Please describe this part of the theory.
Author Response
Dear Reviewer,
Thank You for the review of our submitted manuscript. Individual reviewers had various comments on some parts of the article (even contradictory). We have tried to make changes and improvements in the article based on the compromise and your comments.
Comments and Suggestions for Authors
- Photocurrent is composed of many different linear superposition states, but how to determine the specific states.
Response:
The output current of a multiscan consists of a set of elementary currents of its photodiode cells, and the differentiated output current is a video signal that reflects the intensity of the light relief for each cell of the multiscan being interrogated.
- If there is only one current-to-voltage converter, how to achieve parallel connection of each circuit by layout to in the system.
Response:
The elementary currents of the photodiode cells of the multiscan are summed on its load resistance, then this total current is converted and amplified using a single CVC (current-to-voltage converter) into a voltage and then differentiated to form a video signal from the multiscan in order to build an optoelectronic device based on it.
- Image recognition is easier by Fourier transform, why not use it.
Response:
You can search for the connection between the input radiation distribution Ф(x) and the output video signal using the Fourier transform in the spatial frequency domain. However, the physics mechanism for the formation of the video signal from multiscan is most evident in the considered in the article spatial-temporal region, while the video signal (described by the formula (16)) reflects dual nature of discrete-continuous structure of multiscan: the signal may be discrete (pulse) or analog (continuous) form, depending on the voltage step between cells multiscan. The dual nature of discrete-continuous structure multiscan used in the article in the proposed Vernier discrete-analog method of measuring the parameters of lighting zones on multiscan, which with the aim of increasing the measurement accuracy, the location information of the video pulse is determined relative to the adjacent reference pulses of fixed geometric raster, due to the topology of a discrete structure of multiscan.
- In Figure 4, it can mark the data line, otherwise it is more difficult to determine what each data line is.
Response:
Each data line in Figure 4 is a video signal from a single-line multiscan that reflects the light distribution along its photosensitive surface. Since Figure 4 shows the video signal V (obtained in MathCad) from a single-line multiscan for its first ten photodiode cells (of which cells 2-9 are evenly illuminated) for three cases of decreasing the offset voltage (which corresponds to the step voltage). From Figure 4 it can be seen that in analog mode, the bell-shaped video signals from neighboring cells merge, and the trapezoidal video signal from the multiscan described by expression (16) (Figure 4) becomes completely analogous to the video signal from the solid scanistor.
- The photodetector senses the light intensity. I want to know how it scans the object when the light intensity of the object is the same. Please describe this part of the theory.
Response:
The multiscan determines the intensity of light at each point on its photosensitive surface. To do this, switching of the next discrete cell of the multiscan is performed at the moment when the sawtooth scanning voltage of the survey reaches the voltage level on the cell from the ins. The cell photodiode is switched from the closed state to the open state, and the switching diodes are switched from the open state to the closed state. If the next cell to be interrogated is illuminated, its photodiode and switching diodes form an elementary output current corresponding to the illumination, jn. The total output current In with a common bus of photodiodes of multiscan converted by CVC to voltage, after which it is being differentiated by differentiating amplifier DA and then a video signal V is formed, proportional to the distribution of light along the photosensitive surface of multiscan. It should be noted that the article deals with light zones on a multiscan with uniform illumination.

Reviewer 3 Report
In general, I find the manuscript quite good, it takes up an interesting issue, and the results described may be worth publishing. Unfortunately, the current version of the manuscript has many flaws that need to be corrected before possible publication:
- The paper is difficult to follow; it is written in "strange" language (English must be corrected), and the organization of the manuscript is quite chaotic.
- Abstract is lenghty - it should be clear and concise.
- Introduction is short and superficial - as a result, I do not feel introduced to the issue, just as I do not feel justification for taking it up. Lumped references should be avoided!
- Section 2 - where is the justification for the use of such and not another construction?
- Description of the device given in Fig. 5 is unclear, e.g. what is "AND scheme"? Is it logic gate?
- Where is discussion with literature on she subject? The authors should point out what is their original contribution in the field in order to show novelty of the paper.
In conclusion, I have to say that while the ideas of the paper are certainly not without merit, the overall quality of the manuscript significantly mitigates the enthusiasm. The authors need to rewrite this manuscript with an eye for clear communication.
Author Response
Dear Reviewer,
Thank You for the review of our submitted manuscript. Individual reviewers had various comments on some parts of the article (even contradictory). We have tried to make changes and improvements in the article based on the compromise and your comments.
Comments and Suggestions for Authors
In general, I find the manuscript quite good, it takes up an interesting issue, and the results described may be worth publishing. Unfortunately, the current version of the manuscript has many flaws that need to be corrected before possible publication:
- The paper is difficult to follow; it is written in "strange" language (English must be corrected), and the organization of the manuscript is quite chaotic.
Response:
English in the article has been modified.
- Abstract is lenghty - it should be clear and concise.
Response:
The abstract has been modified and shortened.
- Introduction is short and superficial - as a result, I do not feel introduced to the issue, just as I do not feel justification for taking it up. Lumped references should be avoided!
Response:
The introduction has been modified and extended. The references are grouped with respect to the research area described in this section.
- Section 2 - where is the justification for the use of such and not another construction?
Response:
The justification for the proposed design of an optoelectronic device is a discrete-analog Vernier method for measuring the parameters of the light zone on a multiscan developed and described in the article.
- Description of the device given in Fig. 5 is unclear, e.g. what is "AND scheme"? Is it logic gate?
Response:
Yes, "AND scheme" is a logic gate.
- Where is discussion with literature on she subject? The authors should point out what is their original contribution in the field in order to show novelty of the paper.
Response:
The authors of the article have developed a mathematical model of multiscan in scanistor mode circuit. The proposed new discrete-analog Vernier method of measuring parameters of a light zone on multiscan, which with the aim of increasing the measurement accuracy, the location information of the video pulse is determined relative to the adjacent reference pulses of fixed geometric raster, due to the topology of a discrete structure multiscan. An optoelectronic device is developed based on the proposed Vernier method for measuring the parameters of the light zone on a multiscan.
In conclusion, I have to say that while the ideas of the paper are certainly not without merit, the overall quality of the manuscript significantly mitigates the enthusiasm. The authors need to rewrite this manuscript with an eye for clear communication.
